# Quality of Life and Its Correlates in People Serving Prison Sentences in Penitentiary Institutions

**DOI:** 10.3390/ijerph18041655

**Published:** 2021-02-09

**Authors:** Bartłomiej Skowroński, Elżbieta Talik

**Affiliations:** 1Institute of Social Prevention and Resocialization, University of Warsaw, 00-721 Warsaw, Poland; b.skowronski@uw.edu.pl; 2Department of Social Sciences, John Paul II Catholic University of Lublin, 00-721 Warsaw, Poland

**Keywords:** quality of life, prisoners, prisons, quality of life in prison inmates

## Abstract

Background: The aim of the study was to analyze the determinants of prison inmates’ quality of life (QoL). Methods: 390 men imprisoned in penitentiary institutions were assessed. Data were collected by means of the Sense of Quality of Life Questionnaire (SQLQ), general self-efficacy scale (GSES), resilience assessment scale (RAS-25), social support scale (SSS), intensity of religious attitude scale (IRA), SPI/TPI, and COPE Inventory, measures that have high validity and reliability. All models were specified in a path analysis using Mplus version 8.2. Results: The positive correlates of QoL are: self-efficacy, social support, intensity of religious attitude, trait and state depression, resilience, and the following coping strategies, which are at the same time mediators between the variables mentioned above and QoL: behavioral disengagement, turning to religion, planning, and seeking social support for instrumental reasons. Conclusions: In penitentiary practice, attention should be devoted to depressive individuals, and support should be provided to them in the first place because depressiveness is the strongest negative correlate of important aspects of prisoners’ QoL. All the remaining significant factors, namely: self-efficacy, social support, intensity of religious attitude, and the following coping strategies: turning to religion, planning, and seeking social support for instrumental reasons, should be taken into account in rehabilitation programs.

## 1. Introduction

In the most general terms, quality of life (QoL) is understood as satisfaction with life, “the sense of its meaningfulness, purposefulness, and agency” [1]. Straś-Romanowska [2,3] defines QoL as the contents of life experiences combined with their subjective cognitive and emotional evaluation. Straś-Romanowska’s concept encompasses four QoL dimensions: psychophysical, psychosocial, personal, and metaphysical. The first dimension is understood as the biological basis for the sense of identity, consciousness, volition, and other aspects of human psychological experience [3]. The psychosocial dimension is the sense of security resulting from adaptation to the environment. The personal dimension refers to self-identity, the need for freedom, the need for creative activity, a sense of individuality, self-determination, and self-realization. Finally, metaphysical dimension is human spirituality, with universal and transcendent values (truth, goodness, and beauty) being central to it [3].

Imprisonment may lead to a decrease in inmates’ QoL [4,5]. This process is a result of the deprivation of many important needs—above all, the need for autonomy and freedom and the need for social contact. Prisoners are deprived of good material conditions or high economic status—factors that increase QoL [6]. Prison inmates’ low self-perceived QoL is also associated with low emotional intelligence [6].

Higher QoL is associated with a lower number of conflicts. This was confirmed by Alzúa, Rodríguez, and Villa, who found that QoL was correlated with the level of conflicts in prison [7]. As some authors have noted, research on prisoners’ QoL is needed in order to minimize the harmful consequences of imprisonment. As the goal is to maintain a safe environment for the staff and for prisoners [8], research into the determinants of prison inmates’ QoL seems justified by the need to prevent psychosocial and psychiatric problems.

The associations between prisoners’ QoL and psychopathology are not always direct. As some authors have noted, the following illnesses are typically found in the context of prisons: anxiety, depression, bipolar disorder, psychosis, psychopathy, schizophrenia, and personality disorders [9,10,11]. These illnesses are widespread in prison populations [12], and mentally ill prisoners often return to prison [13]. Psychopathological factors accounted for the majority of the known QoL variance in a sample of Dutch prisoners with mental disorders [14]. The factors that decrease QoL include drug addiction [15] and various kinds of personality disorders, such as antisocial personality disorder [16] and borderline personality disorder [17]. Imprisonment is associated with suicide risk, and factors related to prisoners’ severe suicide attempts include a range of potentially modifiable clinical, psychosocial, and environmental variables [18]. Mental health problems (both current and historical: major depressive symptoms, psychosis, anxiety, and drug misuse disorders) have been identified as factors associated with, and potentially precipitating, near-lethal suicide attempts in prisoners [18]. The comorbidity of disorders is common and significantly associated with near-lethal attempts. Some authors report associations of suicide risk with aggression, impulsivity, hostility, childhood trauma, and hopelessness [18,19] or with low levels of social support and self-esteem [20,21]. On the basis of two long-term research projects, Liebling concluded that there were important differences between those who attempted suicide and other prisoners. Those differences, relating to criminal justice histories and background characteristics, were differences of degree. Individuals who attempted (to commit) suicide had suffered more severe disadvantage, violence, and family problems in their lives and had more frequently come into contact with social services and criminal justice agencies [22]. Important differences were also found in the descriptions of life in prison, which those who had attempted suicide saw as more difficult than other inmates [22]. The number of suicide attempts in Polish prisons was 208 in 2018 and 198 in 2019 (the latter including 3 adolescent prisoners, 39 adults, and 83 adult recidivists). As regards the number of suicide attempts in remand prisons, it was 71 and 2 among those individuals who had committed misdemeanors. The number of suicides committed in Polish prisons was 25 in 2018 and 23 in 2019. It is worth mentioning that the total prisoner population on 31 December 2018, was 72,204, and on 31 December 2019, it was 74,130 inmates [23].

The studies conducted to date have revealed that the variables identified as positive factors for prison inmates’ QoL include optimism, emotional intelligence, future time perspective [6], social support [24], and sociodemographic variables, such as higher education [25]. The variables identified as positive correlates of prisoners’ QoL include those whose positive association with self-perceived QoL has been empirically confirmed in samples other than prisoners, too. For example, self-efficacy, defined as the belief in one’s ability to cope with difficult situations [26], is a significant predictor of self-perceived QoL in parents of children with cerebral palsy [27], in people with multiple sclerosis [28], or in individuals suffering from pain disorders [29]. The most often investigated and at the same time the most significant predictors of QoL are: resilience, being the quality that enables a person to positively adapt to unfavorable conditions [29,30], and social support [31,32,33], whose positive effect on QoL has been found also in a sample of prison inmates [24]. Additionally, in the present study we investigated one variable that had not been analyzed before in the context of QoL: intensity of religious attitude, understood as the certainty and strength of a person’s positive or negative attitude towards God and the supernatural domain [34]. Many researchers have explored the impact of religious practice on prison life and on how prison inmates cope with the dehumanization that can occur in the prison context. Stringer [35] points out that religion allows an individual to survive the loss of freedom and resolve the feelings of guilt and inadequacy while taking personal responsibility for their actions or influences. Moreover, religious affiliation in prison inmates is helpful in modifying behavior and psychological states [36]. A higher level of religiosity is linked to enhanced mental health adjustment and fewer reports of disciplinary confinement [35]. According to Maruna et al. [37], “the conversion narrative can integrate disparate and shameful life events into a coherent, empowering whole, renew prisoners’ sense of their own personal biography, and provide them with hope and a vision for the future” (p. 180). The significance of the intensity of religious attitude for a person’s QoL is indirectly suggested by studies highlighting the importance of religiosity to QoL [38,39,40]. However, despite the scarcity of literature devoted directly to this subject, some studies do exist. It was found that prisoners with high QoL more often chose positive religious coping strategies, whereas those with low QoL more often chose negative ones [41].

The negative correlates selected in the current study were state and trait anxiety, anger, and depression, as defined by Spielberger [42]; their negative association with QoL has been found in numerous studies [43,44,45].

Coping styles can be positive and negative correlates of QoL too. Coping with stress has been found to be significantly and, in most cases, strongly associated with psychological well-being, a variable identified as a concept closely related to QoL [46,47]. Taken together, coping styles (particularly emotion-focused coping) explained as much as 62% of the variance in psychological well-being [48].

Undoubtedly, a number of other factors can be mentioned that potentially affect QoL in prisoners. Discussing the current problems of the Polish prison system, Machel [49] highlighted the low level of employment among prisoners and the need to develop programs offering more educational options, thus giving inmates the opportunity to participate in education. The author also mentioned the need to develop therapeutic programs for prisoners [49].

## 2. Materials and Methods

The aim of the study was to analyze the determinants of QoL in prison inmates. We formulated the following general research problem:

Q1: What are the determinants of QoL in prison inmates?

The specific questions were as follows:

Q2: Which variables are positive correlates of prison inmates’ QoL?

Q3: Which variables are negative correlates of prison inmates’ QoL?

We formulated the following research hypotheses, pertaining to variables not analyzed before in the context of prisoners’ QoL:
**Hypothesis** **1** **(H1).***Self-efficacy, resilience, intensity of religious attitude, and social support are positive correlates of prison inmates’ QoL*.
**Hypothesis** **2** **(H2).***Anger, anxiety, and depression are negative correlates of prison inmates’ QoL*.

In order to test the hypotheses, we conducted a study on a group of 390 men imprisoned in penitentiary institutions, aged 19–68 (M = 35.19, SD = 9.65). The study was conducted in April 2014. The largest number of participants had vocational (26.7%) or elementary education (18.5%), and only 7.7% of the sample was people with higher education. Inmates from big cities (over 150,000 inhabitants) constituted 38.2% of the sample.

The majority of the individuals who took part in the study reported Roman Catholicism (76.4%), and 12.3% reported no religion at all; only 3.8% of the sample was protestant and 3.6% were orthodox. As many as 68% of the participants were believers or strong believers; 17.9% of the sample were believers, and 14.9% were nonbelievers. Most participants (71.3%) engaged in religious practices. Although the sample was not specifically recruited from among prisoners with strong religious belief, 68% of the respondents were prisoners who declared themselves as believers or strong believers.

We intended to include women in our sample, but we received too few completed questionnaires from women.

The study was conducted in correctional facilities in Poland administered by the District Inspectorate of Prison Service in Warsaw: in the Warsaw-Grochów, Warsaw-Białołęka, Warsaw-Mokotów, and Warsaw-Służewiec Remand Prisons and in the Warsaw-Białołęka Penitentiary. Participation in the study was absolutely voluntary. Prisoners were invited to participate in it by a person conducting the research (not by a member of the prison staff). Informed consent was obtained from all participants in the study, and the questionnaires were distributed among those who agreed to participate. The completed questionnaires were collected by the researcher. Thus, the participants remained anonymous to the prison administration.

We used several measures in the study.

The Sense of Quality of Life Questionnaire (SQLQ) by Maria Straś-Romanowska [2] measures global QoL and consists of 60 items (rated on a 4-point Likert scale: strongly disagree, disagree, agree, and strongly agree). The internal consistency of the scale (Cronbach’s alpha) is α = 0.77 for the psychophysical QoL scale, α = 0.71 for the psychosocial QoL scale, α = 0.72 for the personal QoL scale, α = 0.65 for the metaphysical QoL scale, and α = 0.70 for the total score. In this article present an analysis of QoL total score as the sum of all scale scores. The correlations between the SQLQ and other measures (construct validity) were significant and ranged from 0.30 (Psychophysical QoL scale) to 0.53 (Personal QoL scale) [3]. Example items are: “There are more successes than failures in my life” and “I like what I do”.

The general self-efficacy scale (GSES) by Schwarzer, Jerusalem, and Juczyński [50], measuring self-efficacy, is based on Bandura’s concepts of expectations and personal self-efficacy. The measure consists of 10 items (rated on a 4-point Likert scale: not at all true, hardly true, moderately true, and exactly true). Internal consistency was α = 0.85; the test–retest correlation coefficient (with a five-week interval) was 0.78 [50]. Example items are: “I can always manage to solve difficult problems if I try hard enough” and “It is easy for me to stick to my aims and accomplish my goals”.

The resilience assessment scale (SPP-25) by Ogińska-Bulik and Juczyński [51] measures the general level of resilience and its five factors: perseverance and determination in action, openness to new experiences and sense of humor, personal coping skills and tolerance of negative emotions, tolerance of failure and treating life as a challenge, and optimistic approach to life and focus in difficult situations [51]. The measure consists of 25 items, and each of its scales consists of five items (rated on a 5-point Likert scale: strongly agree, agree, undecided, disagree, and strongly disagree). The values of Cronbach’s alpha ranged from 0.67 to 0.75, the test–retest correlation coefficient (with an interval of four weeks) was 0.85. Example items are: “I am open to new experiences” and “I like challenges”.

The social support scale (SWS) was developed by Kmiecik-Baran [52] based on Tardy’s theory [53]. It measures global social support and its four types: informational, instrumental, appraisal, and emotional support (each scale consisting of six true-false items). Cronbach’s alpha coefficient ranged from 0.70 to 0.82 [52]. Example items (referring to supportive people) are: “They warn me when I am in danger” and “They spend a lot of time for me”.

The intensity of religious attitude scale (SIPR) by Prężyna measures the intensity of a person’s attitude towards God and the entire supernatural reality (as understood in the Christian tradition) [34,54]. The author of SIPR assumed that religiosity had three components: cognitive, emotional–motivational, and behavioral. He believed that the object of religious attitude was God (and reality marked by supernaturalism). This means that religiosity manifests itself in the recognition of God as the creator of the world and the ultimate goal of man, in the conscious acceptance of dependence on God, and in the worship of God. The SIPR consists of 30 items (rated on a 7-point Likert scale: completely agree, mostly agree, slightly agree, undecided, slightly disagree, mostly disagree, and completely disagree), 17 of them expressing a positive attitude and 13 expressing a negative attitude. Cronbach’s alpha coefficient was 0.96. The test–retest correlation coefficient (with an interval of seven days) was 0.98 [54]. Example items are: “The world without God is incomprehensible” and “People who love God have found the truth and the greatest good”.

The SPI/TPI by Spielberger [42] comprises two independent parts. The first part (SPI) measures anxiety, depression, curiosity, and anger as emotional states experienced at a particular moment, while the second part (TPI) measures the same emotions as personality traits. Each of the scales consists of 10 items (which makes a total of 80 items, rated on a 4-point Likert scale: almost always, often, sometimes, and never). The values of Cronbach’s alpha ranged from 0.82 to 0.92 for the SPI and from 0.68 to 0.88 for the TPI. Example items measuring the feeling of state anger are: “I feel furious” and “I feel angry”.

The COPE inventory was developed by Carver, Scheier, and Weintraub [55] and adapted into Polish by Juczyński and Ogińska-Bulik [56]. It consists of 60 items representing 15 strategies, four items per strategy (with a 4-point Likert scale: almost always, often, sometimes, and never) The COPE is a multidimensional inventory developed to assess the different coping strategies people use in response to stress. Five scales (active coping, planning, suppression of competing activities, restraint coping, and seeking social support for instrumental reasons) measure problem-focused coping (e.g., “I concentrate my efforts on doing something about it” and “I take additional action to try to get rid of the problem”). The next five scales (seeking social support for emotional reasons, positive reinterpretation and growth, acceptance, denial, and turning to religion) measure emotion-focused coping (e.g., “I discuss my feelings with someone” and “I seek God’s help”). Finally, the COPE inventory also includes three scales that measure coping responses: focus on and venting of emotions, behavioral disengagement, and mental disengagement (e.g., “I get upset and let my emotions out” and “I get upset, and am really aware of it”). The values of Cronbach’s alpha ranged from 0.48 to 0.94, being the lowest for the mental disengagement and active coping strategies and the highest for turning to religion. The test–retest stability coefficients ranged from 0.45 to 0.82. The concurrent validity of the inventory was examined by correlating COPE scores with CISS scores (Coping Inventory for Stressful Situations). A high positive correlation was found between task focus (CISS) and the following variables measured using the COPE inventory: planning (0.70), suppression of competing activities (0.64), and active coping (0.62). A relationship was also found between emotion-focused strategies (CISS), and the use of emotional social support (0.45). Moreover, avoidance style (CISS) correlated positively with mental disengagement (0.62). Self-efficacy (GSES) correlated positively with active coping (0.42) and negatively with behavioral disengagement (−0.43). Active coping strategies correlated with high self-esteem measured using the Rosenberg SES, internal locus of control measured using the Rotter I-E Scale, and anxiety measured using the Spielberger STAI [56]. All the measures used in the present study are suitable for use with a correctional population.

## 3. Results

Table 1 presents the means and standard deviations for the variables.

Raw scores were juxtaposed with the norms specified by the authors of the SQLQ. It should be noted that these norms were set on the basis of research conducted on various age groups: adolescents (*N* = 93), adults (*N* = 73), and seniors (*N* = 55) [57]. Juxtaposing raw scores with the norms led to the conclusion that the mean overall QoL scores fell within the normal range, though it is worth noting that the overall level of QoL verged on the bottom limit of the norm.

The score on the intensity of religious attitude for normalized data [54] showed a low level of this variable. Anxiety, anger, and depression were measured both as states (SPI) and as traits (TPI). In both cases, the mean values were similar. Descriptive statistics for the strategies of coping with stress are presented in Table 2.

The coping strategies least often used by prison inmates were: behavioral disengagement (M = 7.6, SD = 2.68), substance use (M = 7.8, SD = 3.47), denial (M = 7.9, SD = 2.83), and humor (M = 7.9, SD = 2.82).

We performed a correlation analysis and a path analysis to test the hypotheses and to determine the directions of relationships among a set of variables. Table 3 shows Pearson’s product-moment correlation coefficients between the independent variables and QoL.

As is evident from the contents of Table 3, state anxiety, trait anxiety, state anger, trait anger, state depression, and trait depression correlated significantly and negatively with prisoners’ QoL, whereas social support, resilience, intensity of religious attitude, and self-efficacy were significantly and positively correlated with QoL in prison inmates. The strongest correlate was trait depression (*r* = −0.621).

Among the styles of coping with stress, the strongest correlates of prisoners’ quality of life are: planning (*r* = 0.498), positive reinterpretation and growth (*r* = 0.425), active coping (*r* = 0.417), and substance use. Turning to religion and focus on and venting of emotions are not significant; behavioral disengagement (*r* = −0.379) and substance use (*r* = −0.395) correlate negatively with prisoners’ QoL.

In order to get deeper insight into the research results, we used structural equation modeling (SEM). The basic model we constructed included the styles of coping with stress as mediators between independent variables (in particular: self-efficacy, resilience, intensity of religious attitude, social support, depression, anger, and anxiety) and QoL as the dependent variable. In this model, we examined the following coping styles: active coping, planning, positive reinterpretation and growth, substance use, and behavioral disengagement. These coping strategies are the strongest correlates of QoL (see Table 4).

All models were specified in a path analysis using Mplus version 8.2. We used the following goodness-of-fit indices, based on Hu and Bentler’s recommendations: comparative fit index (CFI), Tucker–Lewis index (TLI), root mean square error of approximation (RMSEA), and standardized root mean square residual (SRMR) [58]. The strength of relationships between the study variables was estimated using standardized path coefficients.

According to the traditional SEM fit indices, the tested models provided unacceptable or extremely unacceptable fits to the dataset. This was the starting point for the search for a new model. We tested many different models with QoL predictors as independent variables and QoL as the dependent variable and with different relationships between them. These models were calibrated based on a modified index and theoretical analysis.

We found a new model, which did not include all initial predictors, although the correlations between them and QoL (Pearson’s product-moment correlation coefficients) were significant. The path model is presented in Figure 1.

The final model provided a good fit to the data according to traditional structural equation modeling fit indices, χ^2^ = 41.247, *df* = 17, *p* > 0.05; RMSEA = 0.06 (90 percent C.I. 0.037, 0.084); SRMR = 0.049; CFI = 0.974; TLI = 0.947. The tested model, which is presented together with standardized path coefficients in Figure 1, accounted for significant variance in QoL scores (*R*^2^ = 0.642).

### 3.1. Direct Effects

Significant predictors of QoL are: trait depression (β = −0.274, *p* < 0.001), self-efficacy (β = 0.215, *p* < 0.001), social support (β = 0.211, *p* < 0.001), behavioral disengagement (β = −0.177, *p* < 0.001), planning (β = 0.168, *p* < 0.01), intensity of religious attitude (β = 0.157, *p* < 0.001), resilience (β = 0.114, *p* < 0.01), seeking social support for instrumental reasons (β = −0.098, *p* < 0.05), and turning to religion (β = 0.097, *p* < 0.05) (see Table 5).

### 3.2. Indirect Effects

It can be seen in Figure 1 that social support can indirectly impact QoL through behavioral disengagement (β = −0.039, *p* < 0.01); seeking social support for instrumental reasons (β = −0.017, *p* < 0.05), and turning to religion (β = −0.020, *p* < 0.05). The impact of social support on QoL through seeking social support for instrumental reasons and through turning to religion is nonsignificant; however, the *p*-value is slightly lower than the required statistical significance threshold (*p* = 0.074). The next factor that indirectly impacts QoL is the intensity of religious attitude, related to QoL through turning to religion (β = 0.048, *p* < 0.05), seeking social support for instrumental reasons, and turning to religion (β = −0.006, *p* < 0.05). Depression (as a trait only) impacts QoL through behavioral disengagement (β = −0.038, *p* < 0.01). Finally, resiliency impacts QoL through planning (β = 0.095, *p* < 0.001), seeking social support for instrumental reasons, and planning (β = −0.025, *p* < 0.05).

## 4. Discussion

The aim of the present study was to analyze the determinants of prison inmates’ QoL. As predicted, the positive correlates of QoL are: self-efficacy, social support, intensity of religious attitude, trait and state depression, resilience, and the following coping strategies, which are at the same time mediators between the factors mentioned above and QoL: behavioral disengagement, turning to religion, and planning and seeking social support for instrumental reasons.

Some of the relationships revealed by the path analysis seem surprising and should be explained. First of all, we observed that social support negatively predicts turning to religion, which led to seeking social support for instrumental reasons. However, this relationship does not result in higher QoL in prisoners; it actually reduces their QoL. Perhaps this is due to the nature of the entire institution of prison. It seems that a prisoner who does not find social support in the penitentiary institution turns to religion in difficult situations [59,60] and at the same time looks for support but does not receive it. This may suggest that imprisonment leads to a decrease in inmates’ QoL [4,5], mainly due to the deprivation of many important needs—above all, the need for autonomy and freedom and the need for social contact [6,61,62].

On the other hand, prisoners who apply the strategy of turning to religion also use the behavioral disengagement strategy. Perhaps the observed covariance between turning to religion and behavioral disengagement could be explained by the distinction between personal and impersonal religiosity [63]. Personal religiosity is a type of religious experience associated with activity, commitment, spontaneity, autonomy, a sense of freedom and responsibility, creativity, orientation towards community, allocentricity, openness, and a sense of dignity. Impersonal religiosity, by contrast, is associated with passivity, indifference, rigidity, a sense of compulsion, existential disorientation, egocentrism, selfishness, artificiality, and the isolation of religion from other spheres of life [63]. It is noteworthy that prisoners’ behavioral disengagement is related to depression, which suggests that, probably, prisoners’ religiosity is impersonal rather than personal. This issue requires further exploration, however.

Other interesting relationships occurred between intensity of religious attitude, turning to religion, seeking social support for instrumental reasons, and QoL. Intensity of religious attitude was positively related to turning to religion, which led to seeking social support for instrumental reasons, but this did not result in higher QoL. Perhaps the religiosity of some prisoners is personal and in stressful situations they cope by turning to religion (this results from the intensity of their religious attitude), which induces them to seek support. Unfortunately, they do not receive it. It seems that either the prison institution does not provide prisoners with the support they need, or the support offered is unsuccessful (i.e., does not result in higher QoL), although prisoners seek it.

Self-efficacy seems to be quite a strong predictor of prisoners’ QoL. This is consistent with the findings of other studies, concerning ill people [28,64] and also prison inmates [65]. In the case of prisoners, this factor is associated with positive readaptation of individuals at risk of a criminal career [66].

The present study confirms that another significant positive determinant of prisoners’ QoL is social support [24]. This is understandable in the context of the functioning of prison inmates, for whom one of the priority challenges is to find their place in the prison hierarchy and to adapt to the norms established by the prison subculture [10]. In other words, the more a person feels a member of the prison subculture, the more satisfied his or her needs are, such as the need for belonging, security, and social contact. This also refers to the needs connected with the possibility of making choices and with opportunities to pursue personal goals and interests. Generally, subjectively perceived QoL increases with the amount of appraisal support received. In this context, relations with the prison staff are of importance; research shows that the more positive and supportive they are, the higher is the prisoners’ QoL [67,68].

A significant positive correlate of QoL is the intensity of religious attitude, which directly relates to supernatural reality [34]. Interestingly, its level in the prison inmates examined in the present study was low, which may suggest that they exhibited a confrontational attitude towards God [54]. However, this issue requires further exploration. It is worth stressing, however, that the deeper prisoners’ religiosity became, the greater increase there was in the level of their QoL. These results are consistent with the findings of other studies, highlighting the importance of religiosity for QoL [39,40,41,69,70], effective coping with stress [71], and inmates’ psychosocial functioning [38,65,66].

Consistently with expectations and with the results obtained in other studies [29,30], one of the positive correlates for inmates’ QoL is resilience, the quality that makes it possible for a person to positively adapt in unfavorable conditions. Our research shows that resilience explains prison inmates’ QoL. In other words, the more strongly a person is convinced that they have the skills to cope in difficult situations and the ability to refrain from expressing the negative feelings experienced, the higher is the person’s satisfaction with life and with themselves and the better is also their self-rated psychophysical condition. The ability to inhibit one’s own emotional responses is one of the “unwritten” rules of functioning in the prison subculture, which enables optimal functioning in the situation of imprisonment [72]. The lack of personal skills of coping with negative emotions leads to serious consequences, including personal ones.

A component of resilience is tolerance of failure and treating life as a challenge, which may be interpreted as the ability to come to terms with negative situations and as acceptance of failures instead of agonizing over them or blaming oneself for them [51]. The more a prison inmate adopts this attitude of acceptance of the inevitable though usually just fate, the more easily he or she opens up to higher values, such as good, love, truth, and beauty. The attitude of accepting an unfavorable situation is conducive to acting in accordance with one’s conscience and contributes to the development of morality [52].

The present study reveals that, as expected, the negative correlates of prison inmates’ QoL are state and trait depression, state anger, and state anxiety. Their negative association with QoL has been found in numerous studies [43,44,73], which means the results of the present study are consistent with the results of previous ones. It is worth noting that prison inmates’ temporary and current experience of anxiety, depression, and anger (as states) is associated with a decrease in psychophysical and metaphysical QoL, whereas only trait depression affects the psychosocial, subjective, and global dimensions of prisoners’ QoL. In other words, negative emotions experienced here and now affect the individual’s psychophysical condition: well-being, vitality, and spiritual life, whereas being a depressive person translates into a decrease in general satisfaction with life, with oneself, and with interpersonal relations (the exception in this case is state anger, which also lowers general satisfaction with life), which means it affects areas important to prisoners’ everyday functioning. This gives rise to the suggestion that, in penitentiary practice, attention should be devoted to depressive individuals and that support should be provided to them in the first place because depressiveness—trait depression—is an important negative correlate of important aspects of prison inmates’ QoL [73] and because it is strictly related to suicide risk [74].

Some of the tested strategies of coping with stress predict QoL. The following strategies are mediators: behavioral disengagement, turning to religion, planning, and seeking social support for instrumental reasons. This conclusion is in line with research results on the relationship between coping with stress and QoL [75] and on the relationship between QoL and religious coping [41]. Imprisonment and the ensuing frustration of many needs may naturally lead to the loss of the sense of meaning in life and, consequently, to a decrease in inmates’ subjective well-being. For this reason, it seems important to look for solutions in the penitentiary system that will help prisoners find meaning and develop a belief in the possibility of coping with the situation of imprisonment. Research shows that one of the ways to accomplish this is to provide them with opportunities to name and express their experiences and emotions and with opportunities to pursue their interests [76].

Coping with stress is related to self-efficacy, resilience, religiosity, and the experience of negative emotional states associated with depressive disorders. When an individual is faced with a problem, self-efficacy comes into play and instigates a particular coping strategy that matches the level of the individual’s self-efficacy [55,77]. High self-efficacy has been associated with active coping (e.g., problem-solving and information seeking) and low self-efficacy with passive coping (e.g., avoidance and distraction). The most commonly reported active coping strategy used by individuals with high self-efficacy is problem-focused coping [55,77,78,79]. Resilience correlated positively with task-oriented coping and negatively with disengagement and distraction-oriented coping. Analysis of variance indicated that athletes with high individual resilient qualities scored higher on task-oriented coping, using disengagement and distraction-oriented coping to a lower extent. The results obtained suggest that resilient characteristics in athletes may be associated with the use of potentially more adaptative coping strategies [80]. Epidemiological studies indicate that people frequently rely on religion to cope with life’s stresses. This demonstrates the positive impact of religious coping on well-being [81,82,83]. The experience of negative emotional states associated with depressive disorder narrows the attention span, reduces the ability to engage in flexible and creative thinking, and also reduces adaptive capacity. This limits effective coping in stressful situations in the present and in the future. The shortage of coping resources contributes to the deterioration of QoL, which adversely affects health condition [84].

To sum up, as expected, the positive correlates of various aspects of prison inmates’ QoL are: self-efficacy, resilience, intensity of religious attitude, social support, and some of the coping strategies. As hypothesized, the negative correlates of inmates’ QoL are: anger, anxiety, state depression, and trait depression, which are consistent with the results of previous studies. A novelty of the study is the inclusion of a variable that has not been analyzed before in the context of prison inmates’ QoL or QoL in general: intensity of religious attitude. The limitation of the presented research is its failure to take gender differences into account in analyses due to the absence of women in the sample. Given the results obtained to date, for instance those concerning depression showing that the levels of this variable is higher in women [42,85], women can be expected to differ significantly from men in terms of the type of positive and negative correlates of QoL. This hypothesis can be a point of departure for further research. It would also be an interesting challenge to conduct longitudinal research into the relations between protective and risk factors for QoL in a longer-term perspective and in the context of the type of crimes committed: is it not the case that one of the risk factors increases the likelihood of committing a particular type of punishable offense while a different one protects against relapse into crime? This issue is addressed in studies concerning, for instance, the impact of religiosity on the decrease or increase in criminal tendencies in a particular population [86,87].

The results of the present study may contribute to the construction of programs supporting the development of personal resources: self-efficacy and resilience, which proved to be positive correlates of QoL in prisoners. What is worth noting is the significance of religiosity, which is an important source of QoL for many prison inmates. Seeking social support implies that what should also be taken into account is the system of social relations that, despite obvious limitations, are important to prisoners. When designing prevention programs, by contrast, it is worth considering those variables that turned out to correlate negatively with QoL (state and trait depression, anxiety, and hostility).

Another limitation of the present study is its failure to include some variables pertaining to the prison context, namely: the social climate of prisons, conflicts in prisons, health problems, age at incarceration, length of stay in prison, length of sentence, or even the size of the prison cell. The authors did not obtain all data from prison staff (e.g., the size of prison cells) and the data obtained from the prisoners were incomplete, which made it impossible to carry out analyses in this regard.

## Figures and Tables

**Figure 1 ijerph-18-01655-f001:**
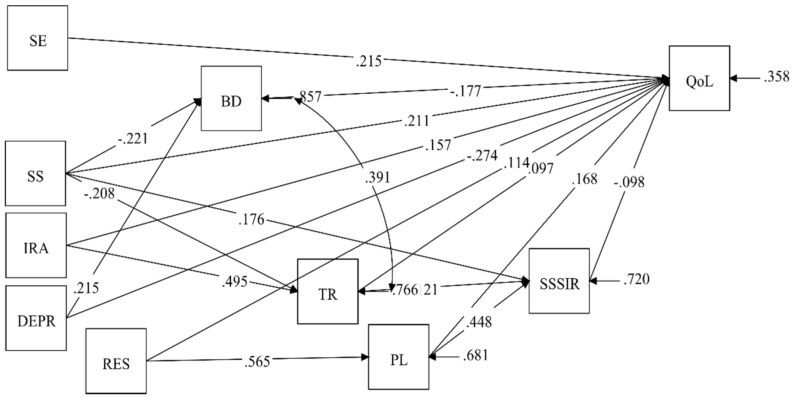
Path model. SE—self-efficacy; SS—social support; IRA—intensity of religious attitude; DEPR—TPI depression; RES—resilience; BD—behavioral disengagement; TR—turning to religion; PL—planning; SSSIR—seeking social support for instrumental reasons; QoL—quality of life.

**Table 1 ijerph-18-01655-t001:** Means and standard deviations for the study variables.

			M	SD	Min.	Max.
Global QoL			2.99	0.37	1.55	3.85
Positive correlates	self-efficacy		30.7	5.55	10	40
	social support		84	14.87	31	116
	general level of resilience		68.7	18.67	0	100
	intensity of religious attitude		88	24.48	20	140
Negative correlates	states	SPI anxiety	21.1	5.58	10	40
		SPI anger	20.2	7.12	10	40
		SPI depression	20.6	5.85	10	40
	traits	TPI anxiety	21.5	4.53	10	37
		TPI anger	21.4	6.43	10	40
		TPI depression	20.2	5.28	10	40

**Table 2 ijerph-18-01655-t002:** Descriptive statistics for strategies of coping with stress (*N* = 390).

	M	SD	Min.	Max.
Active coping	10.7	2.62	4	16
Planning	11.1	2.88	4	16
Seeking social support for instrumental reasons	10.5	2.83	4	16
Seeking social support for emotional reasons	9.8	2.83	4	16
Suppression of competing activities	10.4	2.56	4	16
Turning to religion	8.2	3.34	4	16
Positive reinterpretation and growth	10.5	2.70	4	16
Restraint	9.9	2.42	4	16
Acceptance	9.7	2.69	4	16
Focus on and venting of emotions	9.8	2.59	4	16
Denial	7.9	2.83	4	16
Mental disengagement (Self-distraction)	8.6	2.45	4	16
Behavioral disengagement	7.6	2.68	4	16
Substance use	7.8	3.47	4	16
Humor	7.9	2.82	4	16

**Table 3 ijerph-18-01655-t003:** Pearson’s product-moment correlation coefficients for QoL and its positive and negative correlates.

	Social Support	Resilience	Intensity of Religious Attitude	SPI Anxiety	SPI Anger	SPI Depression	TPI Anxiety	TPI Anger	TPI Depression	Self-Efficacy
QoL	0.565 **	0.585 **	0.385 **	−0.524 **	−0.521 **	−0.605 **	−0.508 **	−0.344 **	−0.621 **	0.489 **
Social support	1	0.398 **	0.261 **	−0.498 **	−0.471 **	−0.482 **	−0.496 **	−0.369 **	−0.501 **	0.283 **
Resilience		1	0.223 **	−0.493 **	−0.392 **	−0.532 **	−0.425 **	−0.231 **	−0.566 **	0.463 **
Intensity of Religious Attitude			1	−0.180 **	−0.172 **	−0.181 **	−0.081	−0.135 **	−0.168 **	0.174 **
SPI Anxiety				1	0.743 **	0.863 **	0.757 **	0.517 **	0.771 **	−0.272 **
SPI Anger					1	0.666 **	0.586 **	0.651 **	0.604 **	−0.225 **
SPI Depression						1	0.740 **	0.467 **	0.808 **	−0.317 **
TPI Anxiety							1	0.590 **	0.771 **	−0.277 **
TPI Anger								1	0.476 **	−0.118 *
TPI Depression									1	−0.286 **
Self-efficacy										1

Note. * *p* < 0.05; ** *p* < 0.01.

**Table 4 ijerph-18-01655-t004:** Pearson’s product-moment correlation coefficients between QoL and coping styles.

	QoL
Active coping	0.417 **
Planning	0.498 **
Seeking social support for instrumental reasons	0.262 **
Seeking social support for emotional reasons	0.328 **
Suppression of competing activities	0.310 **
Turning to religion	ns.
Positive reinterpretation and growth	0.425 **
Restraint	0.172 **
Acceptance	0.135 **
Focus on and venting of emotions	ns.
Denial	−0.327 **
Mental disengagement (Self-distraction)	−0.181 **
Behavioral disengagement	−0.379 **
Substance use	−0.395 **
Humor	−0.242 **

Note. ** *p* < 0.01.

**Table 5 ijerph-18-01655-t005:** Results depicting the relationships observed among the study variables included in the model.

	Estimate	*SE*	Est./*SE*	*R* ^2^
Quality of life				0.642
Self-efficacy	0.215 ***	0.035	6.165	
Social support	0.211 ***	0.038	5.563	
Depression (trait)	−0.274 ***	0.040	−6.786	
Resiliency	0.114 **	0.043	2.626	
Behavioral disengagement	−0.177 ***	0.036	−4.922	
Turning to religion	0.097 *	0.038	2.531	
Planning	0.168 ***	0.042	3.980	
Intensity of religious attitude	0.157 ***	0.037	4.237	
Seeking social support for instrumental reasons	−0.098 **	0.036	−2.702	
Turning to religion				0.234
Intensity of religious attitude	0.495 ***	0.038	12.868	
Social support	−0.208 ***	0.045	−4.610	
Behavioral disengagement				0.143
Social support	−0.221 ***	0.052	−4.266	
Depression (trait)	0.215 ***	0.049	4.352	
Planning				0.319
Resiliency	0.565 ***	0.034	16.396	
Seeking social support for instrumental reasons				0.280
Turning to religion	0.121 **	0.044	2.770	
Planning	0.448 ***	0.043	10.521	
Social support	0.176 ***	0.046	3.831	

Note. * *p* < 0.05; ** *p* < 0.01; *** *p* < 0.001.

## Data Availability

The data presented in this study are available on request from the corresponding author. The data are not publicly available due to privacy.

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
