# Peer review of "Quality of Life and Its Correlates in People Serving Prison Sentences in Penitentiary Institutions"

_ijerph, 2021, doi:10.3390/ijerph18041655_

Round 1

Reviewer 1 Report

The article has presented a lot of aspects in QoL, and it was nice to join in discusion that topic.

As far as my opinion it is worth brightening theoretical basic up. This article presents enought determinants QoL, but only in general level (skim the surface). The first part (introduction) attracted me to think all dereminants out, but reader asks afther more information.  

For instance:

  1.  suicide attampt in prison- we need statistics and comments. In references there are polish language articles, so please to check polish statistic: 2018- 208 occurrances, 2019- 198 in general population about 74 000. Look at gap- young prisoners (3), adult (39) and adult recidivists- 83! Presentation a wide variety of suicide problem in different groups of prisoners must brighten up perception.
  2. familly problems as determinat, but we can’t go over prisoner’s socialization, childhood. That topic is so appealing, because it isn’t only theoretical determinant, it is a part of personal experience. Their story of life makes an impact on QoL and current outlook on life.
  3.  religious strategies in description doesn’t apeal to me. In prison there are musilims, orthodox, buddhists, and a wide range of religious motivations, resistance. Strategies in prison have many different colours. What about different needs of cultural and religious diversity?

Generally speking theoretical basis in insufficient. It goes without saying we need much more information and new references. Please keep in mind that there are a lot of publications in penitnetiary pedagogy. New references (H. Machel, G. Szczygieł, A. Urbanek, Przegląd Więziennictwa Polskiego as Journal) broaden horizons.

On the other hand there are many terms without comment:  mentally ill, mental health, mental disorders. We need venture deep, because sweeping information distorts the result. For instance: How should we interpret the sentence „mentally ill prisoners often return to prison”? Look at polish penal code (seq 25), polish executive penal code (seq 150), comment is nessesary.

By the way references.  Problem of derpivation of needs is explained relevant to one article [6]. It is important aspect in penitnetiary pedagogy. That probem was explorated, because deprivation is a kind of stigma of their life, not only in prison.  

Much as I admire this invention, all pedagogical research were skipped, why?

The next step- methodology.

Research problems and hypotheses matches with research plan. Survey and study take off research plan.

But there are a few questions:

  1. are there any limitation for this model of methodology in prison?,
  2. are there distorted conclusions predictable, because of old-time research (2014)
  3.  what about different type of prisoners (young-adult, adult- adult recidivist), different penal’s models in Poland for instance- therapeutic, corrective? I am certain that isolation, control, deprivation, threat- aren’t thak tactors important for QoL.

Prison inmates is general population, but why specific gropus were skipped?

Author Response

We would like to thank the Reviewer for his thoughtful comments and efforts towards improving our manuscript. In the following, we highlight general concerns of the reviewer and our effort to address these concerns. Here are our responses to the Reviewer's comments.

  • Reviewer’s comment:

“1.       suicide attampt in prison- we need statistics and comments. In references there are polish language articles, so please to check polish statistic: 2018- 208 occurrances, 2019- 198 in general population about 74 000. Look at gap- young prisoners (3), adult (39) and adult recidivists- 83! Presentation a wide variety of suicide problem in different groups of prisoners must brighten up perception”

Our response:

We have inserted the following passage:

"The number of suicide attempts in Polish prisons was 208 in 2018 and 198 in 2019 (the latter including 3 adolescent prisoners, 39 adults, and 83 adult recidivists). As regards the number of suicide attempts in remand prisons, it was 71 and 2 among those individuals who had committed misdemeanours. The number of suicides committed in Polish prisons was 25 in 2018 and 23 in 2019. It is worth mentioning that the total prisoner population on December 31, 2018, was 72,204, and on December 31, 2019, it was 74,130 inmates (Central Board of the Prison Service, 2019)."

  • Reviewer’s comment:

“2.       familly problems as determinat, but we can’t go over prisoner’s socialization, childhood. That topic is so appealing, because it isn’t only theoretical determinant, it is a part of personal experience. Their story of life makes an impact on QoL and current outlook on life”

Our response:

This seems to be an interesting issue, but we only took into account those predictors of QoL that are supported in the literature. We have not found studies investigating these factors (socialization, life experience) in relation to QoL in prisoners.

  • Reviewer’s comment:

“3.       religious strategies in description doesn’t apeal to me. In prison there are musilims, orthodox, buddhists, and a wide range of religious motivations, resistance. Strategies in prison have many different colours. What about different needs of cultural and religious diversity?”

Our response:

We controlled for this variable. Our survey included an item about prisoners’ religion. We have inserted the following passage:

“The majority of the individuals who took part in the study reported Roman Catholicism (76.4%), and 12.3% reported no religion at all; only 3.8% of the sample were Protestant and 3.6% were Orthodox. As many as 68% of the participants were believers or strong believers; 17.9% of the sample were believers, and 14.9% were nonbelievers. Most participants (71.3%) engaged in religious practices.”

The RCOPE, which we used, is a suitable tool for measuring religious coping in the context of the Christian tradition.

  • Reviewer’s comment:

“Generally speking theoretical basis in insufficient. It goes without saying we need much more information and new references. Please keep in mind that there are a lot of publications in penitnetiary pedagogy. New references (H. Machel, G. SzczygieÅ‚, A. Urbanek, PrzeglÄ…d WiÄ™ziennictwa Polskiego as Journal) broaden horizons”.

Our response:

We have inserted the following passage:

“Undoubtedly, a number of other factors can be mentioned that potentially affect the QoL in prisoners. Discussing the current problems of the Polish prison system, Machel (2009) highlighted the low level of employment among convicts and the need to develop programs offering more educational options, thus giving inmates the opportunity to participate in education. The author also mentioned the need to develop therapeutic programs for prisoners (Machel, 2009)”.

  • Reviewer’s comment:

On the other hand there are many terms without comment:  mentally ill, mental health, mental disorders. We need venture deep, because sweeping information distorts the result. For instance: How should we interpret the sentence „mentally ill prisoners often return to prison”? Look at polish penal code (seq 25), polish executive penal code (seq 150), comment is nessesary.

Our response:

What we mean by “mentally ill prisoners” is people suffering from the disorders mentioned in lines 53–54.

  • Reviewer’s comment:

By the way references.  Problem of derpivation of needs is explained relevant to one article [6]. It is important aspect in penitnetiary pedagogy. That probem was explorated, because deprivation is a kind of stigma of their life, not only in prison. 

Our response:

Thank you for this comment! The relationship between QoL and stigma is a very interesting problem. We have been conducting another study on this issue. QoL-related stogma has been researched and found in people with mental illnesses, but probably not yet in prison inmates (see Åšwitaj, Grygiel, Chrostek, Nowak, Wciórka, Anczewska, 2017, Quality of Life Research, 26:2471–2478).

  • Reviewer’s comment:

Much as I admire this invention, all pedagogical research were skipped, why?

Our response:

We don’t know what the Reviewer really means by “pedagogical research.” We have chosen variables that are predictors of QoL, confirmed in the other populations. These factors should be modified in the resocialization process (e.g., development of social support, development of therapeutic programs).

  • Reviewer’s comment:

“1.       are there any limitation for this model of methodology in prison?”

Our response:

We have mentioned some limitations of the present study. One of them is the failure to include some variables associated with the prison context, namely: the social climate of prisons, conflicts in prisons, health problems, age at incarceration, length of stay in prison, length of sentence, or even the size of the prison cell. Another limitation is the fact that we did not obtain all data from prison staff (e.g., the size of prison cells), and the data obtained from the prisoners were incomplete. Moreover, the results of our study refer only to men, not to women. We received too few completed questionnaires from women.

  • Reviewer’s comment:

“2.       are there distorted conclusions predictable, because of old-time research (2014)”

Our response:

We believe our results are stable. We know that the Polish prison system is constantly changing, but in the last 6 years there have been no changes in the penitentiary system that would affect the results of our research.

  • Reviewer’s comment:

“3.       what about different type of prisoners (young-adult, adult- adult recidivist), different penal’s models in Poland for instance- therapeutic, corrective? I am certain that isolation, control, deprivation, threat- aren’t thak tactors important for QoL”.

Our response:

Unfortunately, we did not control for these variables. Thank you for this comment. We intend to take this variable into account in the future research.

Reviewer 2 Report

Abstract

L 14 – no evidence that intensity of religious attitude has positive correlation with quality of life

Introduction

Not clear what the purpose of the paper is/what point it is trying to prove or explore

31-38 the 4 quality of life dimensions are given as though they cannot include anything else and are completely static which seems unlikely. Perhaps needs to be rephrased

39 – I would question if imprisonment does lead to a decrease in quality of life for all – what about people on the streets, those on drugs, those in abusive relationships etc – for many it reduces their quality of life but for others it does not.

52-55 not clear whether it means that people are imprisoned because they are mentally ill or prison induces mental illness. And the reality is much more complex than either of these perspectives.

59-74 strange terminology ‘suicide attempters’ and again it massively oversimplifies the reality and Alison Liebling’s research.

82-83 not clear why the article jumps from prisons to parents of children with cerebral palsy etc.

90-95 there are studies on impacts of religion on those incarcerated – Palgrave Macmillan have a number of books which address this, see researchers such as Kerley, Clear, Sumter, Hewitt et al. I would also dispute the assertion that those with high quality of life see religion as positive whilst those with a low quality see it as negative.

Materials and Methods

116 conducting a study on a group of men – issue with patriarchal intonation of this.

Throughout this section it is unclear how the multiple different scales and tests were combined within one questionnaire or whether each participant was asked to complete multiple questionnaires. No idea how the sample were selected and what their perceived quality of life was, whether there were differences as you would expect in different prisons, different stages of sentences, lengths of sentences, family ties, programmes they were undertaking, etc

Was this research taken as a snapshot in time and if so, to what were the respondents comparing their answers to?

Seems to be no acceptance that prison will impose a significant number of issues on someone’s life but that the factors which can influence their quality of life within the prison system can be significant – acceptance of this more nuanced reality would make the article stronger and have a potentially tangible output.

Discussion

Not clear what the selected correlations are and why they have been chosen

327-336  Again this seems to oversimplify the complexity of the situation – some inmates find solace in religion and I would challenge that it is impersonal religiosity whilst at the same time the level of autonomy is restricted in the penal environment.

The argument seems cyclical and the points about mediators are repeated more than once.

424-430 difficult to follow the leaps between prisoners, athletes and religion

442 Very late stage to note that the sample were male only. This would have been better included in the introduction along with the justification of why that was the case.

There is little in the way of conclusion beyond a list of limitations and exclusions from the project.

Author Response

We would like to thank the Reviewer for his thoughtful comments and efforts towards improving our manuscript. In the following, we highlight general concerns of the reviewer and our effort to address these concerns. Here are our responses to the Reviewer's comments.

  • Reviewer’s comment:

“L 14 – no evidence that intensity of religious attitude has positive correlation with quality of life”

Our response:

The results of our study confirm that there is a positive correlation between intensity of religious attitude and QoL (r = .38, p <.01) (Table 3, p. 6). Also, in our model (Figure 1) intensity of religious attitude is a positive predictor of QoL (β = 0.157).

  • Reviewer’s comment:

“Not clear what the purpose of the paper is/what point it is trying to prove or explore”.

 Our response:

The aim of the study was to analyze factors related to QoL in prison inmates as a specific group. We took into account psychosocial predictors that had not been analyzed before in the context of prisoners’ QoL (e.g., intensity of religious attitude) or had been analyzed rarely (the remaining factors). Moreover, we wanted not only to investigate the significance of a set of QoL predictors but also to determine which of them was the strongest predictor of QoL. Finally, we wanted to investigate the indirect effects of selected predictors on prisoners’ QoL.

  • Reviewer’s comment:

“31-38 the 4 quality of life dimensions are given as though they cannot include anything else and are completely static which seems unlikely. Perhaps needs to be rephrased”

Our response:

Due to its subjective nature, quality of life is a dynamic concept. The examples provided, pertaining to specific dimensions of QoL, do not exhaust the multiplicity of its manifestations. For example, in the psychosocial sphere, also other aspects will be of importance – those significant for an individual’s sense of security in social relations.

  • Reviewer’s comment:

“39 – I would question if imprisonment does lead to a decrease in quality of life for all – what about people on the streets, those on drugs, those in abusive relationships etc – for many it reduces their quality of life but for others it does not”.

Our response:

We have changed sentence as follows:

“Imprisonment may lead to a decrease in inmates’ QoL.”

  • Reviewer’s comment:

“52-55 not clear whether it means that people are imprisoned because they are mentally ill or prison induces mental illness. And the reality is much more complex than either of these perspectives”.

Our response:

In the passage in question (52-55) we refer to studies showing that imprisoned people more often suffer from the disorders mentioned, without explaining the direction of this relationship. In the first sentence we underscore that the associations between prisoners’ QoL and psychopathology are not always direct and clear.

  • Reviewer’s comment:

“59-74 strange terminology ‘suicide attempters’ and again it massively oversimplifies the reality and Alison Liebling’s research”. We have inserted these sentences: “those who attempted suicide” or “Individuals who attempted (to commit) suicide”.

Our response:

This term may seem strange, but it was used by Alison Liebling, and we only cite a fragment of her study that points out differences between individuals making suicide attempts and other prisoners.  At the end of the paragraph we add: “moreover, other important differences were found in suicide attempters’ descriptions of life in prison, which they saw as more difficult” (73-74).

  • Reviewer’s comment:

“82-83 not clear why the article jumps from prisons to parents of children with cerebral palsy etc.”

Our response:

We give examples of positive relationships between the variables selected in the project (e.g., self-efficacy) and QoL in other groups, as no evidence of such relationships in the group investigated in our study (i.e., prisoners) is available.

  • Reviewer’s comment:

“90-95 there are studies on impacts of religion on those incarcerated – Palgrave Macmillan have a number of books which address this, see researchers such as Kerley, Clear, Sumter, Hewitt et al. I would also dispute the assertion that those with high quality of life see religion as positive whilst those with a low quality see it as negative”.

Our response:

Naturally, there are many studies on prisoners’ religiosity, but in our study we have included a variable that relates to a specific dimension of religiosity – namely, intensity of religious attitude, understood as the strength of positive or negative attitude towards God. Thus narrowly defined, religiosity has not been researched in a sample of prisoners. The situation is similar in the case of relations between QoL and the specific dimension of religiosity: positive / negative  religious coping strategies (rather than religiosity per se). Research shows that people whose intensity of religious attitude is more positive more often choose positive religious coping strategies, such as collaborative religious coping, seeking spiritual support from God, etc. (Talik, 2012). As explained earlier, we have cited research results pertaining to other groups because the specifically defined variables (intensity of religious attitude, religious coping strategies) have not been a subject of analyses concerning prisoners.

  • Reviewer’s comment:

“116 conducting a study on a group of men – issue with patriarchal intonation of this”.

Our response:

We intended to include women in our sample, but we received too few completed questionnaires from women.

  • Reviewer’s comment:

“Throughout this section it is unclear how the multiple different scales and tests were combined within one questionnaire or whether each participant was asked to complete multiple questionnaires. No idea how the sample were selected and what their perceived quality of life was, whether there were differences as you would expect in different prisons, different stages of sentences, lengths of sentences, family ties, programmes they were undertaking, etc”.

Our response:

-each participant was asked to complete multiple questionnaires;

-the study was conducted in correctional facilities administrated by the District Inspectorate of Prison Service in Warsaw: in the Warsaw-Grochów, Warsaw-BiaÅ‚oÅ‚Ä™ka, Warsaw-Mokotów, and Warsaw-SÅ‚użewiec Remand Prisons and in the Warsaw-BiaÅ‚oÅ‚Ä™ka Penitentiary; participation in the study was absolutely voluntary;

-prisoners’ perceived quality of life was measured using the Sense of Quality of Life Questionnaire (SQLQ), developed by Maria StraÅ›-Romanowska;

-the present study was not meant to assess the differences in prisoners’ quality of life, which is why we did not perform such analyses but focused on identifying the correlates of QoL.

  • Reviewer’s comment:

“Was this research taken as a snapshot in time and if so, to what were the respondents comparing their answers to?”

Our answer:

The aim of our study was not to analyze differences but to look for correlates of QoL in a sample of prisoners – the analyses were performed only within the study sample. Participants’ scores, form example QoL scores, were compared with the mean scores obtained in normalization studies . Raw scores were juxtaposed with the norms specified by the authors of the SQLQ.

  • Reviewer’s comment:

“Not clear what the selected correlations are and why they have been chosen”.

Our answer:

The choice of variables was based on the analysis of the literaturÄ™ on the determinants of quality of life in prisoners and in other groups. Table 3 presents correlations between quality of life (and its dimensions) and the variables selected as positive or negative correlates of quality of life.

  • Reviewer’s comment:

“327-336  Again this seems to oversimplify the complexity of the situation – some inmates find solace in religion and I would challenge that it is impersonal religiosity whilst at the same time the level of autonomy is restricted in the penal environment”.

Our answer:

We refer to Jaworski’s theory of personal and impersonal religiosity and we cite his understanding of both concepts. It is in this context that we explain the obtained results, treating them rather as a hypothesis for further research (prisoners’ religiosity is probably more impersonal than personal, but this issue requires further exploration).

  • Reviewer’s comment:

“424-430 difficult to follow the leaps between prisoners, athletes and religion”.

Our answer:

Our intention was to show similar relationships in other groups – in this case, athletes.

  • Reviewer’s comment:

“442 Very late stage to note that the sample were male only. This would have been better included in the introduction along with the justification of why that was the case.

Our response:

We have inserted the following passage at the beginning of the Method section:

“We intended to include women in our sample, but we received too few completed questionnaires from women”.

  • Reviewer’s comment:

“There is little in the way of conclusion beyond a list of limitations and exclusions from the project”.

Our response:

We have added a few sentences in the Discussion section:

“The results of the present study may contribute to the construction of programs supporting the development of personal resources: self-efficacy and resilience, which proved to be positive correlates of QoL in prisoners. What is worth noting is the significance of religiosity, which is an important source of QoL for many prison inmates. Seeking social support implies that what should also be taken into account is the system of social relations that, despite obvious limitations, are important to prisoners. When designing prevention programs, by contrast, it is worth considering those variables that turned out to correlate negatively with QoL (state and trait depression, anxiety, and hostility).”

Reviewer 3 Report

I am primarily a qualitative researcher so my expertise on quants methods is limited - therefore I do not feel qualified to comment extensively on the specific methodology employed. 

The quality of the writing could be improved - there were numerous syntax/grammar errors, and I found the frequent use of very short sentences rather jarring to read, especially in the introductory sections. 

The original contribution of this research was not clearly explained/set out - this needed to be set out in the introductory section. How does it differ, exactly, from earlier research? What gaps does it fill? I think the research is interesting and potential original, but this was not clearly explained. Perhaps the authors could focus on the religiosity angle and examine/interrogate it more thoroughly, as that seemed to be the particularly original aspects of their findings? Otherwise, it is well-known that QoL in prison is poor in numerous ways - the authors need to be explicitly and forensically clear about the novelty of their findings. The research is interesting and original, but was perhaps under-played in this article. 

There is quite a lot of qualitative work on religiosity in prisons and coping, and this perhaps could be used to expand/develop the discussion of the quants findings here, and how they might different across populations of different countries (perhaps some are more religious than others!) - for example, the US work of Stringer (2009), Spalek, Maruna et al. in the UK

It would be good to say who/which body gave the ethical approval for this research within the methods section - this was implied at the end where an 'institutional review board' is mentioned, but it is not clear whether this was ethical review specifically, though I suspect it was. 

I hope these comments are not discouraging and I wish the authors the best in refining this very interesting piece further. 

Author Response

We would like to thank the Reviewer for his thoughtful comments and efforts towards improving our manuscript. In the following, we highlight general concerns of the reviewer and our effort to address these concerns. Here are our responses to the Reviewer's comments.

  • Reviewer’s comment:

“The quality of the writing could be improved - there were numerous syntax/grammar errors, and I found the frequent use of very short sentences rather jarring to read, especially in the introductory sections”.

Our response:

Introduction and whole article has been proofread.

  • Reviewer’s comment:

“The original contribution of this research was not clearly explained/set out - this needed to be set out in the introductory section. How does it differ, exactly, from earlier research? What gaps does it fill? I think the research is interesting and potential original, but this was not clearly explained. Perhaps the authors could focus on the religiosity angle and examine/interrogate it more thoroughly, as that seemed to be the particularly original aspects of their findings? Otherwise, it is well-known that QoL in prison is poor in numerous ways - the authors need to be explicitly and forensically clear about the novelty of their findings. The research is interesting and original, but was perhaps under-played in this article”.

Our response:

The aim of the study was to analyse factors related to QoL in prison inmates as a specific group. We took into account psychosocial predictors that had not been analyzed before in the context of prisoners’ QoL context (e.g., intensity of religious attitude) or had been analyzed rarely (the remaining factors). Moreover, we wanted not only to investigate the significance of a set of QoL predictors but also to determine which of them was the strongest predictor of QoL. Additionally, we wanted to investigate the indirect effects of selected predictors on prisoners’ QoL.

  • Reviewer’s comment:

“There is quite a lot of qualitative work on religiosity in prisons and coping, and this perhaps could be used to expand/develop the discussion of the quants findings here, and how they might different across populations of different countries (perhaps some are more religious than others!) - for example, the US work of Stringer (2009), Spalek, Maruna et al. in the UK

Our response:

We have inserted the following passage:

“Many researchers have explored the impact of religious practice on prison life and on how prison inmates cope with the dehumanization that can occur in the prison context. Stringer (2009) points out that religion allows an individual to survive the loss of freedom and resolve the feelings of guilt and inadequacy while taking personal responsibility for their actions or influences. Moreover, religious affiliation in prison inmates is helpful in modifying behavior and psychological states (Clear & Sumter, 2002). A higher level of religiosity is linked to enhanced mental health adjustment and fewer reports of disciplinary confinement (Stringer, 2009). According to Maruna et al. (2006), “the conversion narrative can integrate disparate and shameful life events into a coherent, empowering whole, renew prisoners’ sense of their own personal biography, and provide them with hope and a vision for the future” (p. 180)”

  • Reviewer’s comment:

“It would be good to say who/which body gave the ethical approval for this research within the methods section - this was implied at the end where an 'institutional review board' is mentioned, but it is not clear whether this was ethical review specifically, though I suspect it was”.

Our response:

Yes, the approval granted by Institutional Review Board of The Faculty of Education of the Cardinal Stefan Wyszyński University in Warsaw and Research Committee of Cardinal Stefan Wyszyński University in Warsaw (2014), concerned ethical standards.

Round 2

Reviewer 2 Report

L75-81 is the entire study based in Poland?if so it should be made clear at the outset

L123 convicts is a problematic term

L144-148 did you specifically recruit those with strong religion? 

Methods section would benefit from being related to the study more - currently it reads as a list of different tests but with little cohesion

L355 there is a lot of existing research on the impacts of loss of autonomy and social support reducing quality of life in prison

L399 it seems that you are arguing religion is the most important influence on quality of life but I am not convinced that is the case based on the overall research base which exists on prison experiences

L430-434 I am not sure that this needs to be stated, I think the link between depression and suicide for those in and outside of prison is already widely known

L490-492 I would think that avoiding depression, hostility and anxiety would already be expected in any form of prison programming?

Author Response

We are sending corrected version of our article. We would like to again thank the Reviewer for the comments. In line 154 we have confirmed that the entire study was in Poland; in line 123 we have inserted “prisoners” instead “convicts”. In turn, in lines  L149-151 we have inserted some details about group recruiting. To support Reviewer’s suggestion, in line 355 we have added some literature about impacts of loss of autonomy and social support reducing quality of life in prison. We would like to explain that we don’t conclude that religion is the most important predictor of QoL, but, as our study has revealed, is statistically significant predictor. Other details of our responses are presented in table below.

Reviewer's comment:

L75-81 is the entire study based in Poland? if so it should be made clear at the outset

Our response:

We have inserted the words highlighted in green (L154):

“The study was conducted in correctional facilities in Poland, administered by the District Inspectorate of Prison Service in Warsaw: in the Warsaw-Grochów, Warsaw-BiaÅ‚oÅ‚Ä™ka, Warsaw-Mokotów, and Warsaw-SÅ‚użewiec Remand Prisons and in the Warsaw-BiaÅ‚oÅ‚Ä™ka Penitentiary”. 

Reviewer's comment:

L123 convicts is a problematic term

Our response:

We have inserted “prisoners” instead “convicts” (L123).

“Undoubtedly, a number of other factors can be mentioned that potentially affect QoL in prisoners. Discussing the current problems of the Polish prison system, Machel [49] highlighted the low level of employment among prisoners and the need to develop programs offering more educational options, thus giving inmates the opportunity to participate in education. The author also mentioned the need to develop therapeutic programs for prisoners [49]”.

Reviewer's comment:

L144-148 did you specifically recruit those with strong religion? 

Our response:

No, we did not. We have added the following passage to make this clear (L149-151):

Although the sample was not specifically recruited from among prisoners with strong religious belief, 68% of the respondents were prisoners who declared themselves as believers or strong believers”. 

Reviewer's comment:

Methods section would benefit from being related to the study more - currently it reads as a list of different tests but with little cohesion

Our response:

We looked for different correlates of QoL in prison inmates, hence the variety of methods. We wanted to test which correlates were the strongest among the investigated set of variables. 

Reviewer's comment:

L355 there is a lot of existing research on the impacts of loss of autonomy and social support reducing quality of life in prison

Our response:

To support that conclusion, we have added some literature:

Ashkar, P. J., & Kenny, D. T. (2008). Views from the inside: Young offenders' subjective experiences of incarceration. International Journal of Offender Therapy and Comparative Criminology, 52, 584–597. doi:10.1177/0306624X08314181

Van der Kaap-Deeder, J., Audenaert, E., Vandevelde, S., Soenens, B., Van Mastrigt, S., Mabbe, E., Vansteenkiste, M. (2017). Choosing when choices are limited: The role of perceived afforded choice and autonomy in prisoners’ well-being. Law and Human Behavior, 41(6), 567–78

Moreover , we have deleted the sentence: “This issue requires further research”.

Reviewer's comment:

L399 it seems that you are arguing religion is the most important influence on quality of life but I am not convinced that is the case based on the overall research base which exists on prison experiences

Our response:

We did not want to argue that religion was the most important Qol predictor in prisoners, but that it was a significant factor related to QoL. We wanted to highlight this conclusion because there are no religious programs in prison work in the Polish penitentiary system. Religiosity understood as a personal resource is often overlooked in rehabilitation programs in Polish prisons.

Reviewer's comment:

L430-434 I am not sure that this needs to be stated, I think the link between depression and suicide for those in and outside of prison is already widely known

Our response:

We would like to believe that this conclusion is widely known to the Polish prison service, but unfortunately we are not sure about that.

Reviewer's comment:

L490-492 I would think that avoiding depression, hostility and anxiety would already be expected in any form of prison programming?

Our response:

We know that in the Polish penitentiary system there are no special systematic programs targeted at the reduction of depression, hostility, and anxiety in prisoners. These goals can be found in psychiatric care, but we are aware of no prevention programs specifically aimed at that. That is why we underline this issue.